# A Novel ALDH2 Activator AD-9308 Improves Diastolic and Systolic Myocardial Functions in Streptozotocin-Induced Diabetic Mice

**DOI:** 10.3390/antiox10030450

**Published:** 2021-03-13

**Authors:** Hsiao-Lin Lee, Siow-Wey Hee, Chin-Feng Hsuan, Wenjin Yang, Jing-Yong Huang, Ya-Ling Lin, Chih-Neng Hsu, Juey-Jen Hwang, Shiau-Mei Chen, Zhi-Zhong Ding, Tung-Yuan Lee, Yu-Chiao Lin, Feng-Chiao Tsai, Wei-Lun Su, Li-Yun Chueh, Meng-Lun Hsieh, Che-Hong Chen, Daria Mochly-Rosen, Yi-Cheng Chang, Lee-Ming Chuang

**Affiliations:** 1Department of Internal Medicine, National Taiwan University Hospital, Taipei 100225, Taiwan; leehsiaolin@ntu.edu.tw (H.-L.L.); d91448003@ntu.edu.tw (S.-W.H.); jingyonghuang@ntu.edu.tw (J.-Y.H.); s920449@yahoo.com.tw (Y.-L.L.); jueyhwang@ntu.edu.tw (J.-J.H.); smcs7133@ntu.edu.tw (S.-M.C.); tsaifc@ntu.edu.tw (F.-C.T.); weilunsu0310@ntu.edu.tw (W.-L.S.); r04455002@ntu.edu.tw (M.-L.H.); 2Division of Cardiology, Department of Internal Medicine, E-Da Hospital, Kaohsiung 824410, Taiwan; calvin34@isu.edu.tw; 3Division of Cardiology, Department of Internal Medicine, E-Da Dachang Hospital, Kaohsiung 82445, Taiwan; 4School of Medicine, College of Medicine, I-Shou University, Kaohsiung 840203, Taiwan; 5Foresee Pharmaceuticals, Co., Ltd., Taipei 11560, Taiwan; wenjin.yang@foreseepharma.com; 6Department of Internal Medicine, National Taiwan University Hospital, Yunlin Branch, Yunlin 64041, Taiwan; Y00677@ms1.ylh.gov.tw; 7Graduate Institute of Medical Genomics and Proteomics, National Taiwan University, Taipei 10055, Taiwan; r07455006@ntu.edu.tw (Z.-Z.D.); r08455003@ntu.edu.tw (T.-Y.L.); 8Department of Pharmacology, National Taiwan University, Taipei 100233, Taiwan; d05443004@ntu.edu.tw (Y.-C.L.); r09455005@ntu.edu.tw (L.-Y.C.); 9Department of Chemical and Systems Biology, Stanford University School of Medicine, Stanford, CA 94305, USA; chehong@stanford.edu (C.-H.C.); mochly@stanford.edu (D.M.-R.); 10Institute of Biomedical Sciences, Academia Sinica, Taipei 115024, Taiwan; 11Graduate Institute of Molecular Medicine, National Taiwan University, Taipei 100233, Taiwan; 12Graduate Institute of Clinical Medicine, National Taiwan University, Taipei 100233, Taiwan

**Keywords:** diabetic cardiomyopathy, 4-hydroxy-2-nonenal, mitochondrial aldehyde dehydrogenase 2, AD-9308

## Abstract

Diabetes mellitus has reached epidemic proportion worldwide. One of the diabetic complications is cardiomyopathy, characterized by early left ventricular (LV) diastolic dysfunction, followed by development of systolic dysfunction and ventricular dilation at a late stage. The pathogenesis is multifactorial, and there is no effective treatment yet. In recent years, 4-hydroxy-2-nonenal (4-HNE), a toxic aldehyde generated from lipid peroxidation, is implicated in the pathogenesis of cardiovascular diseases. Its high bioreactivity toward proteins results in cellular damage. Mitochondrial aldehyde dehydrogenase 2 (ALDH2) is the major enzyme that detoxifies 4-HNE. The development of small-molecule ALDH2 activator provides an opportunity for treating diabetic cardiomyopathy. This study found that AD-9308, a water-soluble andhighly selective ALDH2 activator, can improve LV diastolic and systolic functions, and wall remodeling in streptozotocin-induced diabetic mice. AD-9308 treatment dose-dependently lowered serum 4-HNE levels and 4-HNE protein adducts in cardiac tissue from diabetic mice, accompanied with ameliorated myocardial fibrosis, inflammation, and apoptosis. Improvements of mitochondrial functions, sarco/endoplasmic reticulumcalcium handling and autophagy regulation were also observed in diabetic mice with AD-9308 treatment. In conclusion, ADLH2 activation effectively ameliorated diabetic cardiomyopathy, which may be mediated through detoxification of 4-HNE. Our findings highlighted the therapeutic potential of ALDH2 activation for treating diabetic cardiomyopathy.

## 1. Introduction

Diabetes mellitus (DM) is one of the biggest and rapidly increasing health issues in the 21st century, resulting primarily from sedentary lifestyle and high-calorie diets. The worldwide diabetes population in 2019 was estimated to be 463 million and was expected to reach 578 million by 2030 and 700 million by 2045 [1]. The progression of diabetes is usually accompanied with complications, such as cardiovascular diseases, neuropathy, nephropathy, and retinopathy [2].

Relationship between DM and cardiovascular diseases is well established. DM is a major risk factor for atherosclerotic cardiovascular diseases. In addition, the incidence of heart failure in diabetic patient is also comparatively high [3,4]. Development and progression of heart failure in diabetic patients, without coronary artery, valvular heart disease, or hypertension, was first described in 1972 [5]. This form of heart failure was termed as “diabetic cardiomyopathy”. Initially, diabetic cardiomyopathy is characterized by subclinical changes in structure and function, including myocardial fibrosis and stiffness, and impaired myocardial relaxation. With progression of diabetic cardiomyopathy, left ventricular (LV) remodeling and advanced diastolic dysfunction develop, and it evolves to systolic dysfunction eventually [6,7].

The mechanism behind diabetic cardiomyopathy is complex and multifactorial. Previous studies reported that hyperglycemia (causing glucotoxicity) and elevated fatty acid levels (causing lipotoxicity) in diabetes caused formation of reactive oxygen species, increased inflammation andmyocardial fibrosis, impaired mitochondrial function, and deranged calcium homeostasis [7,8,9,10,11,12].In diabetic hearts, proinflammatory cytokines including interleukin 1β (IL-1β), IL-6, and tumor necrosis factor-α (TNF-α) expression levels first increased [13,14,15,16], followed by activation of transforming growth factor β (TGF-β) and connective tissue growth factor (CTGF) that inducesfibrosis in diabetic cardiomyopathy [8,9,10,11,12,13,14,15,16,17]. Moreover, mitochondrial damage and decreased sarco/endoplasmic reticulum (SR/ER) Ca^2+^ release were observed in diabetic hearts [18,19,20,21].

In DM, lipid peroxidation, induced by hyperglycemia-mediated oxidative stress, generates reactive aldehydes [22]. These aldehydes form covalent adducts with DNA and proteins in tissues, resulting in protein dysfunction, alteration of intracellular signaling, and organelle damage [22,23,24]. 4-hydroxy-2-nonenal (4-HNE) generated from lipid peroxidation when polyunsaturated fatty acids in bilayer cell membrane are attacked by reactive oxygen species, is one of most studied bioreactive aldehydes [22]. Both serum and tissue levels of 4-HNE were increased under diabetic condition, making it as a potential biomarker of diabetic complications [24,25,26]. Studies have shown the association of 4-HNE with heart diseases, such as atherosclerosis, myocardial ischemic injury, ventricular hypertrophy, and cardiomyopathy [27,28]. Animal studies also showed increased 4-HNE or 4-HNE protein adducts in myocardial tissues, causing cardiac damage in diabetic rats and mice [6,27,29,30,31]. Therefore, elevation of 4-HNE may contribute to structural and functional abnormalities in diabetic hearts.

HNE can be detoxified by adduction of the C-3 electrophilic center with reduced glutathione (GSH) by glutathione-S-transferases, oxidation of the aldehyde group by aldehyde dehydrogenases (ALDH) to form 4-hydroxy-2-nonenoic acid (HNEAcid), and reduction of the aldehyde group to an alcohol by aldo-keto-reductases or alcohol dehydrogenases to form 1,4-dihydroxynonene [32]. Aldehyde dehydrogenase 2 (ALDH2), located in mitochondria, is one of the major enzymes for detoxifying 4-HNE, which in turn protects the heart from oxidative stress damage [28,33]. The expression level and enzymatic activity of ALDH2 were reduced in diabetic hearts. Overexpression of ALDH2 was reported to prevent cardiac dysfunction in streptozotocin (STZ)-induced diabetic mice. In addition, a small-molecule ALDH2 activator, Alda-1, was shown to enhance ALDH2 activity and attenuate myocardial injury caused by ischemia-reperfusion [8,28]. The development of small-molecule ALDH2 activators promises a potential treatment for diabetic cardiomyopathy.

AD-9308 is a water-soluble and highly selective ALDH2 activator that is more potent than the prototype drug Alda-1 [34]. In this study, STZ-induced diabetic mice were treated with AD-9308 to examine whether activation of ALDH2 could reverse diabetic cardiomyopathy. Our findings showed that AD-9308 treatment amended both diastolic and systolic cardiac dysfunctions, and reversed ventricular wall remodeling through reducing myocardial fibrosis, inflammation, apoptosis, mitochondrial damage, and improving mitochondrial respiration and calcium homeostasis in diabetic mice.

## 2. Materials and Methods

### 2.1. Animals

Experiments were performed on C57BL6/J mice according to National Ethical guidelines and were approved by the Institutional Animal Care and Use Committee of the Medical College of National Taiwan University (Ethical approval number: IACUC 20200046), which is accredited by the Association for Assessment and Accreditation of Laboratory Animal Care International (AAALAC). Hyperglycemia was induced in 8-week-old mice by intraperitoneal (i.p.) injection of STZ (40 mg/kg/day) for one week. Mice were given 0 (water vehicle), 60, and 180 mg/kg/day of AD-9308 by daily oral gavage since the age of 10 weeks (Figure 1a). The chemical structure of AD-9308 was shown in previous literature [34]. No anti-diabetic therapy was used. The non-diabetic control mice were given vehicles (water) by daily oral gavage.

At the end of experiments, mice were sacrificed by CO_2_ euthanasia and then the heart was harvested. A portion of fresh heart tissue was fixed with 4% formaldehyde for histopathological study, and another small piece was soaked in REzolTM C&T reagent (Protech Technology Enterprise Co., Ltd., Taipei, Taiwan) for gene expression assay. The remaining portion of heart was stored in liquid nitrogen for other assays.

### 2.2. Echocardiography

At 5 months post-STZ induction of diabetes, STZ-induced diabetic mice with or without AD-9308 treatment and non-diabetic control mice received echocardiographic examination. Echocardiography was performed using a high frequency ultrasound imaging system Prospect T1 (S-Sharp Corporation, Taipei, Taiwan) with a 40 MHz transducer. Mice were anesthetized with isoflurane (3–4% for induction and 1–2% for maintenance) during examination. Two-dimensional M-mode echocardiography was performed in the parasternal long-axis view for measuring the parameters of the LV geometry, including LV internal dimension at end diastole and systole (LVIDd and LVIDs), end-diastolic interventricular septal thickness (IVSd) and LV posterior wall thickness (LVPWd). LV endocardial fractional shortening was calculated as (LVIDd − LVIDs)/LVIDd × 100%. LV end diastolic volume (EDV) was calculated as [7.0/(2.4 + LVIDd)] × LVIDd^3^ and LV end systolic volume (ESV) was calculated as [7.0/(2.4 + LVIDs)] × LVIDs^3^. Stroke volume was then derived as EDV−ESV, and ejection fraction as (EDV − ESV)/EDV × 100%. LV mass was calculated as 1.05 × [(IVSd + LVIDd + LVPWd)^3^ − LVIDd^3^]. Early diastolic mitral inflow velocity (E), late diastolic mitral inflow velocity (A), deceleration time of E wave, and isovolumic relaxation time were acquired from the apical 4-chamber view using pulsed-wave Doppler, and the E/A ratio was calculated. Myocardial velocities including peak systolic (s), early (e’) and late (a’) diastolic mitral annular velocities were measured through the apical 4-chamber view using tissue Doppler, and the e’/a’ ratio was calculated. The E/e’ ratio was applied to estimate the LV filling pressure. Each parameter was quantified with the average of measurements obtained from three consecutive cardiac cycles.However, E wave and A wave of some mice are missing in some mice due to technical difficulty.

### 2.3. Hematoxylin and Eosin (H&E) Staining

The mice heart from each group were soaked in 4% formalin, dehydrated through graded ethanol and embedded in paraffin wax. The paraffin-embedded tissue blocks were cut into 0.2 ìm-thick paraffin sections. The tissue sections were deparaffinized by immersing in xyleneand rehydrated. Sections were dyed with hematoxylin and eosin and then rinsed with water. Dried sections were mounted and were taken photomicrographs using Olympus BX51 microscope combined with Olympus DP72 camera and CellSens Standard 1.14 software (Olympus, Hamburg, Germany).

### 2.4. Serum 4-HNE Level Measurement

Blood was collected from mice fasted for 4 h, and then was spun at 7000× *g* for 10 min to obtain serum. Serum 4-HNE levels were measured using ELISA kits (cat. No. EEL-M2677, Elabscience, Wuhan, China) and VICTOR Multilabel plate reader (PerkinElmer, West Berlin, NJ, USA).

### 2.5. ImmunoblottingAnalysis

Total proteins from cardiac tissues were separated using 10% or 15% SDS-PAGE and then transferred to the PVDF membrane. The primary antibodies used in this study are listed in Appendix A. The signals developed with ECL reagent (Millipore, Burlington, MA, USA) were captured by MultiGel-21 (Top Bio, Taipei, Taiwan).

### 2.6. ALDH2 Enzymatic Activity Assays

ALDH2 activity was measured by monitoring the production rate of NADH/min at 340 nm and 25 °C. Enzyme activity was assayed in 100 µL of reaction mixtures containing 50 mM Na_4_P_2_O_7_ (pH 9.5), 0.01% BSA, 2.5 mM NAD^+^, 100 µM 4-HNE, and protein lysate from heart tissue, and then was monitoredwith the change in absorbance at 340 nm for 5 min by spectrophotometer (Infinite^®^ M Plex, Tecan Trading AG, Zurich, Switzerland). Relative ALDH2 activity was compared to ALDH2 activity of non-diabetic control mice.

### 2.7. Gene Expression Analysis

RNA was extracted from frozen cardiac tissues using the REzolTM C&T reagent (Protech Technology Enterprise, Taipei, Taiwan) following the manufacturer’s instructions. The RNA was reverse-transcribed using RevertAid RT Reverse Transcription kit (Thermo Fisher Scientific Inc., Waltham, MA, USA). Real-time qPCR of interest gene was performed with HD 2X SYBR GREEN qPCR MIX (Hong Da Life Science, Taipei, Taiwan) using the Applied Biosystems QuantStudio 7 Flex Real-Time qPCR System and Applied Biosystems Sequence Detection Systems (QuantStudio Software v1.3) software (Applied Biosystems, Waltham, MA, USA). Ribosomal 18S was used as the endogenous control. The primers used in this study are listed in Appendix A. The target gene relative to the 18S reference gene was analyzed using the comparative ÄCt method.

### 2.8. Immunohistochemistry (IHC) Staining

Formalin-fixed heart tissue was paraffin-embedded and sectioned (0.5 ìm thickness). Sections were stained using anti-α-smooth muscle actin (α-Sma, 1:500), anti-Collagen IV (Col IV, 1:500) or inducible nitric oxide synthase (iNOS, 1:200), and images were captured using Olympus BX51 microscope combined with Olympus DP72 camera and CellSens Standard 1.14 software (Olympus, Hamburg, Germany) or using TissueFAXS system (TissueGnostics, Vienna, Austria).

### 2.9. Cell Culture and Treatment

H9c2 cardiomyoblasts were purchased from the Bioresource Collection and Research Center (BCRC, Hsinchu, Taiwan), and then cultured in Dulbecco’s modified essential medium (DMEM) supplemented with 10% fetal bovine serum (Gibco, Brooklyn, NY, USA), 2 mM glutamine, and 100 units/mLof penicillin and streptomycin. Cellular environment was maintained at 5% CO_2_ at 37 °C incubator. H9c2 cells were pre-treated with AD-9308 for 1 h and then incubated in medium containing fatty-acid-free bovine serum albumin (BSA)-conjugated palmitate (0.1 mM) and high D-glucose (33 mM) for 24–72 h.

### 2.10. Cell Counting Kit (CCK)-8 Assay

H9c2 cardiomyoblasts were seeded at a density of 6 × 10^3^ cells per well into a 96-well microplate and incubated for 24 h. Following treatment, the cell viability was determined using CCK-8 assay (Dojindo Molecular Technologies, Kumamoto, Japan) according to the manufacturer’s protocol. The absorbance value was detected at 450 nm using a microplate reader. The absorbance was normalized to cells incubated in control medium, which was taken as 100.

### 2.11. Measurement of ATP Level

ATP level in cells was measured using ViaLight™ plus kit (Lonza, Basel, Switzerland) according to the manufacturer’s protocol. Briefly, cells were incubated with lysis buffer for 10 min and the substrate was added for 2 min. The absorbance value was detected using SpectraMax M5 ELISA plate reader (Molecular Devices, Silicon Valley, CA, USA).

### 2.12. NF-kB Activity Assay

To measure the activity of NF-κB, cells were harvested after treatment, and nuclear lysates were extracted using the Nuclear Extraction kit (Abcam, Cambridge, UK), according to the manufacturer’s protocol. The NF-κB activity in nuclear extract lysates was detected using the NF-kB transcription factor assay kit (Abcam, Cambridge, UK) according to the recommended experimental protocol.

### 2.13. Subcellular Fractionation

After stimulation with indicated concentration of AD-9308, H9c2 cells were trypsinized, transferred to 15 mL tube, and washed with cold PBS. The protein was further fractionated using NE-PER Nuclear and Cytoplasmic Extraction Reagents (Thermo Scientific, Waltham, MA, USA) according to the manufacturer’s instructions.

### 2.14. Ca^2+^ Measurement

Real-time intracellular levels of calcium were measured with a cell-permeable free-Ca^2+^ fluorescent dye, Fura-2-AM, by a ratiometric imaging technique. After removal of culture media, H9c2 cells were loaded with 2 μM Fura-2-AM in ExtraCellular buffer (ECB: 125 mM NaCl, 5 mMKCl, 1.5 mM MgCl_2_, 10 mM glucose, and 1.5 mM CaCl_2_, 20 mM HEPES pH 7.4) containing 1 mM probenecid and 0.04% F127 for 30 min at room temperature and then were washed twice with ECB.The detection of SR/ER Ca^2+^ content, 2 μM thapsigargin was added to block Ca^2+^ uptake into SR/ER by inhibiting sarco/endoplasmic reticulum Ca^2+^-ATPase2 (SERCA2), and 1.5 mM EGTA were added to chelate extracellular Ca^2+^. Fura-2-AM binds with intracellular Ca^2+^ of H9c2 cells was detected using a fluorescence microscopy (Nikon Eclipse Ti microscope) by alternately exciting the cells at 340 and 380 nm every 300 ms. Fluorescence intensities of cells were analyzed with the customized MATLAB script. Intracellular levels of calcium are present as F340/F380 ratio of Fura-2-AM.

### 2.15. Mitochondrial Respiratory Chain Complex Activity Assay

The mitochondrial complex spectrophotometric assays were carried out using published protocols [35]. Complex I and complex II activities were spectrophotometrically measured by examining the decrease in absorbance due to the reduction of 2,6-dichlorophenolindophenol (DCPIP) at 600 nm. Complex III and complex IV-specific activities were measured by monitoring the reduction of oxidized cytochrome C and oxidation of reduced cytochrome C at 550 nm, respectively. Citrate synthase was spectrophotometrically measured by recording the increase in absorbance due to the reaction between 5′,5′-Dithiobis 2-nitrobenzoic acid (DTNB), and CoA-SH to form TNB^2−^ at 412 nm. Relative mitochondrial complex activity was determined by comparison with complex activity of non-diabetic control mice.

### 2.16. Measurements of Mitochondrial Functions

The metabolic activities of cells were determined by measuring the oxygen consumption rate (OCR; indicative of mitochondrial oxidative respiration), using an XF24 extracellular flux analyzer (Seahorse Bioscience, North Billerica, MA, USA). H9c2 cells (6000 cells/well) were grown on Seahorse Bioscience Tissue Culture plates (Seahorse Bioscience, North Billerica, MA, USA). One hour prior to recording, the cell medium was replaced by DMEM containing 25 mM glucose, 5 mM pyruvate, and 2 mM glutamine. Basal OCR (five measurements) was then measured using the Seahorse XF24-3 analyzer after equilibration and calibration according to the manufacturer’s instructions.

### 2.17. Statistical Analysis

Data were expressed as mean ± SEM. One-way ANOVA with trend test was employed to calculate *p*-for-trend among ordinal groups (Prism 8, GraphPad, San Diego, CA, USA). A *p*-value < 0.05 was considered as statistically significant.

## 3. Results

### 3.1. ALDH2 Activator, AD-9308, Ameliorated Diastolic Dysfunction in STZ-Induced Diabetic Mice

Hyperglycemia was induced in 8-week-old C57BL6/J mice by STZ and mice were given 0 (water vehicle), 60 and 180 mg/kg/day of AD-9308 by daily oral gavage since the age of 10 weeks (Figure 1a). As expected, diabetic mice had lower body weight (Appendix A) and higher fasting blood glucose level (Appendix A) than non-diabetic control mice.

Representative tracings of wave velocities obtained using pulse-wave and tissue Doppler are shown in Figure 1b. As can be seen, all diastolic parameters including E/A ratio, e’ wave, E/e’ ratio, e’/a’ ratio, E wave deceleration time, and isovolumic relaxation time were impaired in STZ-induced diabetic mice, but showed dose-dependent improvement after treatment with AD-9308 (E/A ratio: *p*-for-trend = 0.0180, e’: *p*-for-trend = 0.0002, E/e’: *p*-for-trend = 0.0202, e’/a’: *p*-for-trend = 0.0019, E wave deceleration time: *p*-for-trend = 0.0621 and isovolumic relaxation time: *p*-for-trend = 0.0107; Figure 1c–h). These findings revealed that diabetic related LV diastolic dysfunction could be restored by AD-9308 treatment.

### 3.2. AD-9308 Ameliorated Systolic Dysfunction in STZ-Induced Diabetic Mice

Representative tracings of M-mode echocardiography are shown in Figure 2a. STZ-induced diabetic mice showed decreased fractional shortening, ejection fraction, stroke volume, and cardiac output. Systolic myocardial velocity, s wave (a systolic parameter measured by tissue Doppler), was also reduced in diabetic mice. Treatment of diabetic mice with AD-9308 improved all these systolic functions in a dose-dependent manner (fractional shortening: *p*-for-trend = 0.0055, ejection fraction: *p*-for-trend = 0.0042, stroke volume: *p*-for-trend = 0.0006, cardiac output: *p*-for-trend = 0.0004, and s wave: *p*-for-trend = 0.0167, Figure 2b–f).

### 3.3. AD-9308 Partially Ameliorated LV Structural Changes in STZ-Induced Diabetic Mice

As for LV morphology, M-mode echocardiography showed that AD-9308 treatment was associated with lower reductions of IVSd in a dose-dependent manner (*p*-for-trend = 0.0022) in diabetic mice in comparison with non-diabetic control mice (Figure 3a). The reduction of LV mass in diabetic mice was also mitigated after AD-9308 treatment (*p*-for-trend = 0.0015; Figure 3b). However, other characteristics of heart geometry showed no significant difference between diabetic mice and non-diabetic control mice or AD-9308 treated diabetic mice (Figure 3c–g). Histological examination of myocardium revealed no significant difference in cardiomyocyte size between diabetic mice and non-diabetic control mice or AD-9308-treated diabetic mice (Figure 3h and Appendix A).

### 3.4. AD-9308 Treatment Decreased 4-HNE Level and Enhanced ALDH2 Activity

AD-9308 drastically reduced serum 4-HNE level (Figure 4a) and 4-HNE protein adducts (Figure 4b) in cardiac tissue from diabetic mice. Consistent with previous results [33,36], untreated diabetic mice had lower ALDH2 activity when compared with diabetic mice treated with AD-9308 and non-diabetic mice (Figure 4c). However, there was no difference in ALDH2 protein expression levels among groups (Appendix A). 4-HNE has been shown to trigger oxidative stress. Immunoblots revealed that AD-9308 treatment in diabetic mice lowered the expression of heme oxygenase 1 (Ho-1), a protein whose expression is induced in response to oxidative stress (Figure 4d). These data indicate that AD-9308 treatment in diabetic mice was effective in promoting the detoxification of 4-HNE through enhancing ALDH2 activity and alleviation of oxidative stress.

### 3.5. AD-9308 Treatment Reduced Fibrosis of Diabetic Hearts

Fibrosis and inflammation were reported to contribute to ventricular stiffness and dysfunction of diabetic cardiomyopathy [6]. AD-9308 treatment significantly reduced the expressions of fibrosis biomarkers including *Tgf-β1* (*p*-for-trend < 0.0001), *Ctgf* (*p*-for-trend < 0.0001), fibroblast-specific protein 1 (*Fsp1*; *p*-for-trend = 0.0026), periostin (*Postn*; *p*-for-trend = 0.0046), fibronectin (*Fn-1*; *p*-for-trend = 0.0001) and *Tgf-β2* (*p*-for-trend = 0.0004) in a dose-dependent manner (Figure 5a–f) measured by RT-qPCR. On IHC stained sections, AD-9308 treatment decreased α-Sma and Col IV deposition (Figure 5g,h). The *α-Sma* and *ColIV* mRNA expression measured by RT-qPCR confirmed the results of IHC staining (*p*-for-trend = 0.0367 and 0.0113, respectively) (Figure 5i,j). These results showed that AD-9308 treatment reduced myocardial fibrosis.

### 3.6. AD-9308 Treatment Attenuated Inflammation and Apoptosis in Diabetic Hearts

Apoptosis is a distinct feature of diabetic cardiomyopathy and strongly associated with inflammation. AD-9308 treatment reduced dose-dependently the mRNA expression of inflammatory markers including *Il-1β* (*p*-for-trend = 0.0010), *Il-6* (*p*-for-trend = 0.0002), interferon gamma (*Infγ*; *p*-for-trend = 0.0054), monocyte chemoattractant protein 1 (*Mcp-1*; *p*-for-trend = 0.0047), serum amyloid P-component (*Sap*; *p*-for-trend = 0.0005), and *Tnf-α* (*p*-for-trend < 0.0001) in the heart of diabetic mice (Figure 6a–f). Examining the protein levels of inflammation and apoptosis markers showed that AD-9308 treatment increased B-cell lymphoma 2 (Bcl-2) level, and reduced iNOS, Bcl-2-associated X protein (Bax) and cleaved caspase 3 (Casp3) levels, indicating attenuated inflammation and apoptosis in diabetic hearts (Figure 6g).

The effect of AD-9308 treatment on cell apoptosis under diabetic condition was further explored using the H9c2 cardiomyoblast cell line. H9c2 cells were treated with a high level of glucose (33 mM) combined with a high level of palmitate (0.1 mM) to mimic the high-glucose and high-palmitate serum levels of uncontrolled diabetic patients. High-glucose and high-palmitate treatment significantly reduced H9c2 cell viability and ATP production after 72 h when compared with the controls (Figure 6h,i). Concordantly, pretreatment of AD-9308 exhibited elevated Bcl-2 expression, reduced Bax, and cleaved Casp3 protein expression in comparison with the controls (Figure 6j). These data indicated that AD-9308 preserved cell viability by inhibiting apoptosis of cardiac myoblasts.

NF-κB is the major transcription factor responsible for regulating various pro-inflammatory cytokines. NF-κB is normally bound with its inhibitory factor IκB and sequestered in the cytosol [8]. Upon stimulation, IκB is degraded and dissociated from the inactive cytoplasmic complex, thus facilitating the translocation of the active subunit NF-κB p65 into nuclear fraction and further transactivation of downstream genes. Our results showed that AD-9308 treatment increased the stability of IκBα protein in a dose-dependent manner. In addition, immunoblotting showed a decrease in NF-κB p65 nuclear translocation and reduced P65 reporter activity when treating diabetic mice with AD-9308 (Figure 6l,m). Altogether, our findings suggested that AD-9308 treatment attenuated myocardial inflammation and apoptosis.

### 3.7. AD-9308 Treatment Improved Mitochondrial Functions, Suppressed Autophagy and Calcium Handling in Diabetic Mice

Damaged mitochondrial functions in diabetic mice were previously reported [37]. Our study found a significant decrease in activities of complex II and complex III of mitochondrial electron transfer chain of the heart in diabetic mice when compared with non-diabetic control mice and AD-9308-treated diabetic mice (Figure 7a). To detect the rate of mitochondrial respiration, oxygen respiration rate (OCR) was measured using H9c2 cells. Under high-glucose and high-palmitate culture condition, AD-9308 treatment dose-dependently increased spare respiratory capacity, basal and maximal respiration rates, and ATP production in comparison with non-treated cells (Figure 7b,c).

Furthermore, diabetic cardiomyopathy is characterized by reduced SR/ER Ca^2+^ content which may influences myocardial contractility [38,39]. Therefore, we measured SR/ER Ca^2+^ content in H9c2 cells and the effect of AD-9308 on SR/ER Ca^2+^ content. Cells were first incubated in Fura-2-AM, a cell-permeable fluorescent dye, which binds to cytosolic free Ca^2+^. Figure 7d showed the initial basal level of cytosolic free Ca^2+^.

Then, thapsigargin, a SERCA inhibitor, was added to block Ca^2+^ uptake into SR/ER. Therefore, SR Ca^2+^ were released from into cytosol, causing a transient surge in cytosolic free Ca^2+^ level, which returned to normal levels after 10 min of exposure. (Figure 7d). The peak value of thapsigargin-induced SR/ER Ca^2+^ release to cytosol was used as an index for SR/ER Ca^2+^ content. Under high-glucose and high-palmitate culture condition, the SR/ER Ca^2+^ content in H9c2 cells was reduced. AD-9308 treatment rescued the SR/ER Ca^2+^ content in a dose-dependent manner. Figure 7e showed the fluorescent signal of cytosolic Ca^2+^ level released from SR/ER corresponding to the signal peak in Figure 7d.

Oxidative stress in diabetic cardiomyopathy could result in damage to cellular organelles [40]. Recent studies showed that mitochondrial fusion/fission dynamics play an important role in response to mitochondrial damage [41,42,43,44]. Diabetic mice were found to exhibit higher protein levels of Drp1 (dynamin-related protein 1) and lower protein levels of Opa1 (Optic atrophy 1) when compared with non-diabetic control mice, indicating enhanced mitochondrial fission and suppressed fusion in diabetic heart, which was reversed by AD-9308 treatment (Figure 7f). The immunoblotting of Opa1 showed 2 major isoforms, long Opa1 forms (L-Opa1) and short forms (S-Opa1). The protein expression levels of L-Opa1showed no difference among diabetic mice, diabetic mice with AD-9308 treatment and non-diabetic control mice. However, the protein expression levels of S-Opa1showed an increased accumulation in diabetic mice when compared with diabetic mice with AD-9308 treatment and non-diabetic control mice. This imbalanced Opa1 processing was reported to cause heart failure in mice [45]. The autophagy system is a degradation pathway for cells to turn over organelles, which is required for maintaining normal cardiac functions [46,47]. The autophagy markers, Beclin 1, and microtubule-associated proteins 1A/1B light chain 3A and 3B (LC3A/B) II showed increased protein expression levels in untreated diabetic heart, implying activation of autophagy, possibly related to injured organelles in diabetic hearts. AD-9308 treatment lowered these makers of autophagy, suggesting that AD-9308 might ameliorate organelle damage (Figure 7f).

These above-mentioned findings evidenced impaired mitochondrial respiration, and mitochondrial dynamic and calcium handling in diabetic mice could be reversed by AD-9308.

## 4. Discussion

This study demonstrated that AD-9308 treatment protected diabetic mice from heart failure through restoration of ALDH2 activity and reduction of the 4-HNE level in STZ-induced diabetic cardiomyopathy. AD-9308 treatment ameliorated both diastolic and systolic dysfunctions and reversed ventricular wall remodeling of diabetic hearts. AD-9308 treatment reduced fibrosis, inflammation, and apoptosis, improved mitochondrial respiration and calcium handling, and reduced autophagy in cardiac tissues of diabetic mice.

Recent studies have reported biological roles of ALDH2 in ischemic heart, diabetic cardiomyopathy, and cardiac aging [28,48,49]. Increased oxidative stress from elevated toxic aldehydes has been implicated as the key mediator of these cardiac impairments. Chronic hyperglycemia and elevated fatty acid in diabetes induce oxidative stress in diabetic hearts [50,51]. 4-HNE is a toxic aldehyde generated from peroxidation of polyunsaturated fatty acids of cell membrane when attacked by oxidative stress. 4-HNE is highly bioreactive toward proteins by forming covalent adducts, thus causing cellular damage [23,31]. 4-HNE is metabolized by aldehyde dehydrogenases to non-toxic HNEAcid. Indeed, diabetic patients were found to have elevated 4-HNE serum levels and increased 4-HNE protein adducts in the myocardium [52,53]. In our STZ-induced diabetic mice model, 4-HNE level and 4-HNE protein adducts were elevated while ALDH2 activity was lowered in diabetic hearts. We also demonstrated reduction of ejection fraction, fractional shortening, and stroke volume, decrease in E/A ratio, increase in E/e’ ratio, and ventricular wall thinning, compatible with systolic and diastolic cardiac dysfunctions, and ventricular remodeling in diabetic mice as previously reported [6,53,54,55]. Administration of AD-9308 restored ALDH2 activity in cardiac tissues and significantly lowered 4-HNE levels, accompanied with reversal of diastolic and systolic functions as well as ventricular remodeling in a dose-dependent manner.

The present results were consistent with previous findings showing that ALDH2 prevented hyperglycemia-induced cellular dysfunctions in diabetic mice [37,56]. During disease progression of diabetes, ALDH2 activity reduced while the 4-HNE level increased [57]. Increased ALDH2 activity could reduce 4-HNE protein adducts [58,59]. An administration of AD-9308 significantly lowered Ho-1 protein expression, a stress protein induced upon oxidative stress, indicating amelioration of oxidative stress in diabetic micetreated with AD-9308.

Fibrosis is known as the first stage of diabetic cardiomyopathy [60]. Fibrosis biomarkers, including TGF-β1, TGF-β2, and CTGF as fibrosismarkers, and FSP1, POSTEN, and FN-1 as tissue remodeling markers have been systematically examined [61,62]. We found that the expressions of these genes were all increased in diabetic hearts. Moreover, IHC showed increased collagen disposition in the myocardium of STZ-induced diabetic mice. As inflammatory response was reported to increase cardiac oxidative stress and induce fibrosis in diabetic mice [48,63], inflammatory gene expression in our model was measured. We foundthat increased mRNA expressions of several pro-inflammatory genes including *Il-1β* and *Il-6*, *Mcp-1*, *Tnf-α* and *Infγ*, as well as an inflammation functional predecessor *Spa* in diabetic mice, in agreement with previous results [54,64,65].

Inflammation was reported to stimulate apoptosis in diabetic cardiomyopathy [66]. In general, the rate of apoptosis in the myocardium is very low (0.001 to 0.002%) but becomes elevated in diabetic patients. Increased apoptosis causes myocardial dysfunctions [67]. Increase in Casp3, a central executioner of apoptosis, was observed in LV tissue obtained from the failing human hearts [68]. Consistently, we found increased Caspase-3, increased Bcl-2 (an inhibitor of apoptosis), and reduced Bax (an activator of apoptosis) expression in diabetic heart, which could be as reversed by AD-9308 treatment, consistent with previous studies [58,69,70,71].

Furthermore, previous results showed reduction of mitochondrial complex activities in diabetic mice. Similar mitochondrial dysfunction was also recapitulated in our high-glucose and high-palmitate treated cardiac myoblast cell line model. 4-HNE was reported to dampen the electron transport chainof cardiomyocytes from humans and rats with heart failure [72,73,74] and was found to form adducts with mitochondrial complex II protein in diabetic hearts [75]. On the other hand, ALDH2 activation was reported to protect mitochondrial functions of the myocardium and inhibit ALDH2-aggravated mitochondrial impairment indiabetic rats [33,36]. In addition, our findings showed an impairment of Ca^2+^ handling in H9c2 cells under high-glucose and high palmitate culture condition, which could be partially rescued by AD-9308 treatment. Our results were in line with previous findings showing a decrease in SR/ER Ca^2+^ content in cardiomyopathy in STZ-induced diabetic rats [39,76].

STZ-induced diabetic mice showed enhanced autophagy in diabetic cardiomyocytes. Since glucose and fatty acids cannot be effectively taken up in untreated diabetic myocardium, energy deficit and cellular starvation activates autophagy [77,78,79,80]. In addition, autophagy is also triggered by organelle damage. We found increased autophagy marker expression in diabetic hearts, which is reduced by AD-9308 treatment.

Treatment of diabetic cardiomyopathy remains an unmet medical need. In earlier studies on diabetic cardiomyopathy, chronic metformin treatment was reported to restore autophagic activity and inhibit cardiomyocyte apoptosis in OVE26 diabetic mice [78]. Administration of neuregulin-1 (rhNRG-1) in STZ-induced diabetic rats reduced cell apoptosis and fibrosis as well as ameliorated hypertension [70]. Inhibition of mitochondrial fission with Drp1 inhibitor melatonin was also shown to suppress oxidative stress and alleviate mitochondrial dysfunction and cardiac systolic dysfunction in diabetic mice [41]. However, few of them were successfully translated into clinically approved therapy. AD-9308 is a novel, highly potent, selective, and water-soluble small-molecule ALDH2 activator, which has passed phase 1 human clinical trials with acceptable safety and tolerability in healthy subjects. Our findings may be translated into clinical trials and eventually clinical use soon.

Our study has some limitations. First, ALDH2 is ubiquitously expressed. However, we only explore myocardium, leaving other organs unexplored. Indeed, 4-HNE has been shown to oxidize LDL and promote oxLDL accumulation in atherosclerotic plaques in vessels [81,82]. 4-HNE also induced ER stress in endothelial cells [83]. ER stress exerts a negative effect on endothelial cell stability [84]. Endothelial dysfunction may increase arterial stiffness and resistance and cause myocardial hypertrophy. Therefore, the effect of ALDH2 activation might be through a secondary effect from other cells in addition to the myocardium and the cardiac myocytes as characterized in this study.

## 5. Conclusions

Diabetic cardiomyopathy induced through hyperglycemia and oxidative stress is one of the most severe complications in the progression of diabetes. The present study shows for the first time that ALDH2 activator AD-9308 improved cardiac diastolic and systolic functions as well as ventricular wall remodeling in STZ-induced diabetic mice. The results suggest that restoration of ALDH2 activity by AD-9308 can reduce oxidative stress through detoxification of 4-HNE, leading to suppressed fibrosis, inflammation, apoptosis, and autophagy, and improved mitochondrial functions and calcium dynamics in diabetic cardiomyocytes (Figure 8). Our findings highlighted a new therapeutic approach to treat diabetic cardiomyopathy through scavenging toxic aldehydes.

## Figures and Tables

**Figure 1 antioxidants-10-00450-f001:**
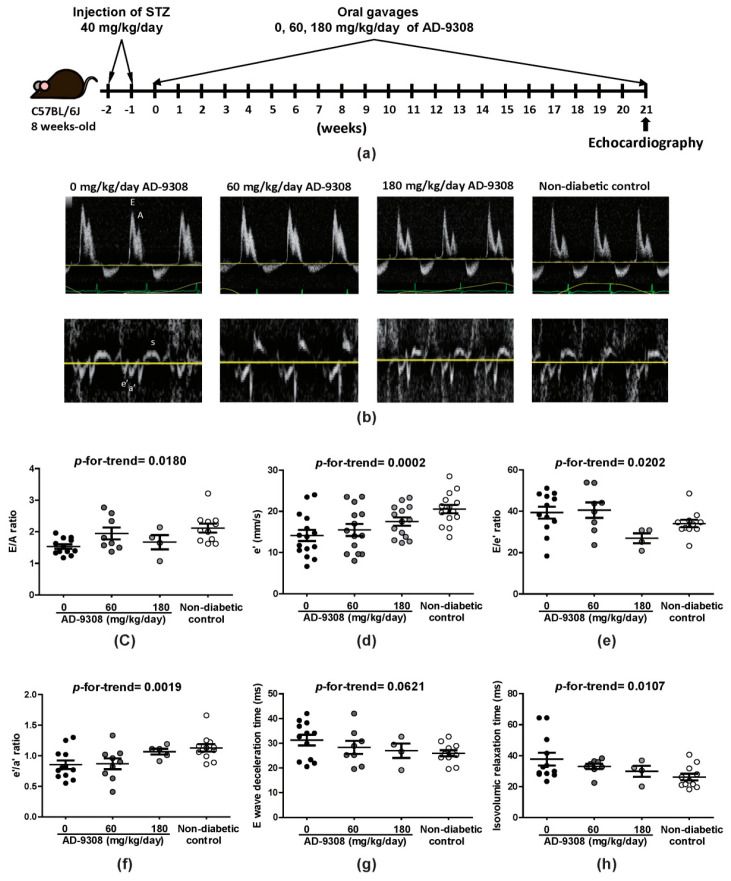
AD-9308 treatment ameliorated left ventricular diastolic dysfunction in streptozotocin (STZ)-induced diabetic mice. (**a**) flow chart illustrating the experimental design, (**b**) representative pulse wave and tissue Doppler echocardiograms. The (**c**) mean E/A ratio, (**d**) e’ wave, (**e**) E/e’ ratio, (**f**) e’/a’ ratio, (**g**) E wave deceleration time and (**h**) isovolumic relaxation time were measured in STZ-induced diabetic mice treated with 0, 60, 180 mg/kg/day of AD-9308 by oral gavages and non-diabetic control mice. Data are presented as mean ± SEM (*n* = 14–16 per group). *p*-for-trend was used to test thelinear trend.

**Figure 2 antioxidants-10-00450-f002:**
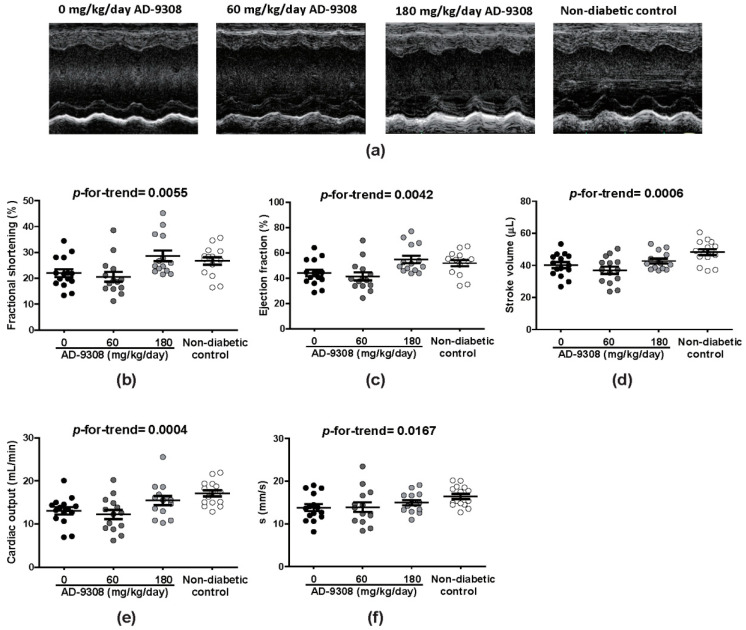
AD-9308 treatment improved left ventricular (LV) systolic dysfunction in streptozotocin (STZ)-induced diabetic mice. (**a**) representative M-mode echocardiograms. The mean (**b**) LV fraction shortening, (**c**) LV ejection fraction, (**d**) stroke volume, (**e**) cardiac output, and (**f**) s wave were measured in STZ-induced diabetic mice treated with 0, 60, 180 mg/kg/day of AD-9308 by oral gavages and non-diabetic control mice. Data are presented as mean ± SEM (*n* = 14–16 per group). *p*-for-trend was used to test the linear trend.

**Figure 3 antioxidants-10-00450-f003:**
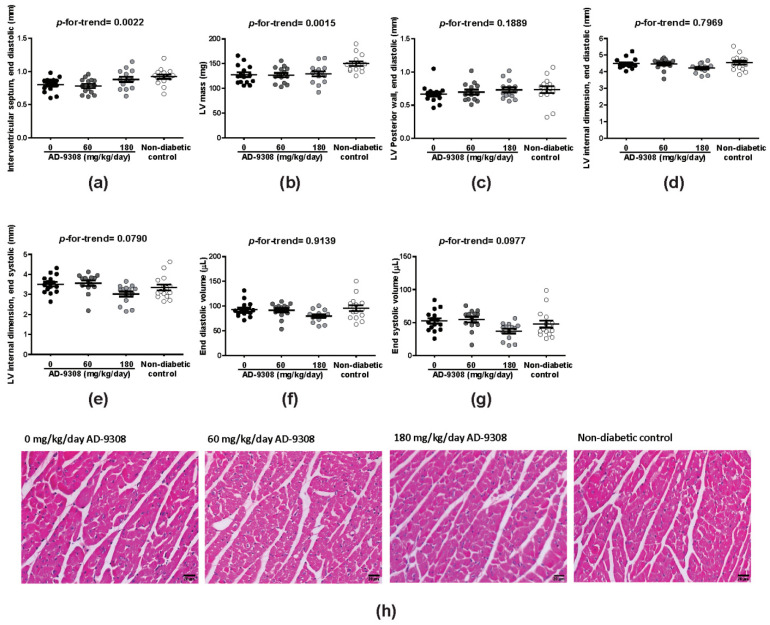
AD-9308 partially normalized cardiac morphology in streptozotocin (STZ)-induced diabetic mice. The mean (**a**) interventricular septum thickness at end diastole, (**b**) leftventricular (LV) mass, (**c**) LV posterior wall thickness at end diastole, (**d**) LV internal dimension at end diastole, (**e**) LV internal dimension at end systole, (**f**) LV end diastolic volume and (**g**) LV end systolic volume were measured in STZ-induced diabetic mice treated with 0, 60, 180 mg/kg/day of AD-9308 by oral gavages andnon-diabetic control mice, (**h**) representative hemotoxylin and eosin staining of heart sections from each group. Data are presented as mean ± SEM (*n* = 14–16 per group). *p*-for-trend was used to test the linear trend. Scale bar, 20 μm.

**Figure 4 antioxidants-10-00450-f004:**
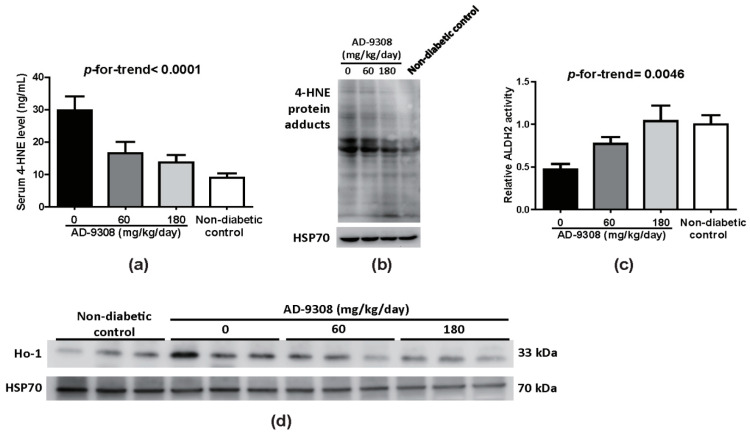
AD-9308 treatment reduced 4-hydroxy-2-nonenal (4-HNE) levels and enhanced aldehyde dehydrogenase 2 (ALDH2) activities to protect streptozotocin (STZ)-induceddiabetic mice against oxidative stress. (**a**) Serum 4-HNE levels, (**b**) 4-HNE protein adducts, (**c**) ALDH2 activities, and (**d**) heme oxygenase-1 (Ho-1) protein expressions of cardiac tissues were measured in STZ-induced diabetic mice treated with 0, 60, 180 mg/kg/day of AD-9308 by oral gavages andnon-diabetic control mice. Data are presented as mean ± SEM (*n* = 6–12 per group). *p*-for-trend was used to test the linear trend.

**Figure 5 antioxidants-10-00450-f005:**
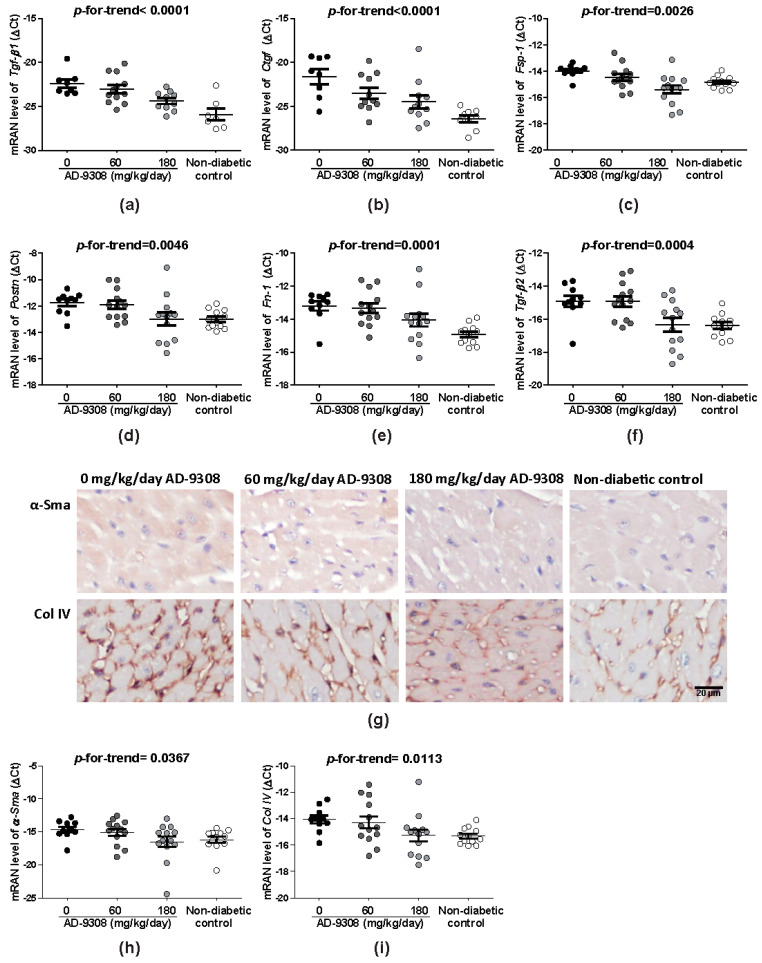
AD-9308 reduced cardiac fibrosis in streptozotocin (STZ)-induced diabetic mice. The mean mRNA levels of fibrosis markers, (**a**) transforming growth factor β1 (*Tgf-β1*), (**b**) connective tissue growth factor (*Ctgf*), (**c**) fibroblast-specific protein 1 (*Fsp1*), (**d**) periostin (*Postn*), (**e**) fibronectin (*Fn-1*), and (**f**) *Tgf-β2*, and (**g**) representative immunohistochemical staining of anti-α-smooth muscle actin (α-Sma)and anti-Collagen IV (Col IV) and mRNA levels of (**h**) *α-Sma* and (**i**) *Col IV* were detected in cardiac tissuesfrom STZ-induced diabetic mice treated with 0, 60, 180 mg/kg/day of AD-9308 by oral gavages and non-diabetic control mice. Data are presented as mean ± SEM (*n* = 14–16 per group). *p*-for-trend was used to test the linear trend. Scale bar, 20 μm.

**Figure 6 antioxidants-10-00450-f006:**
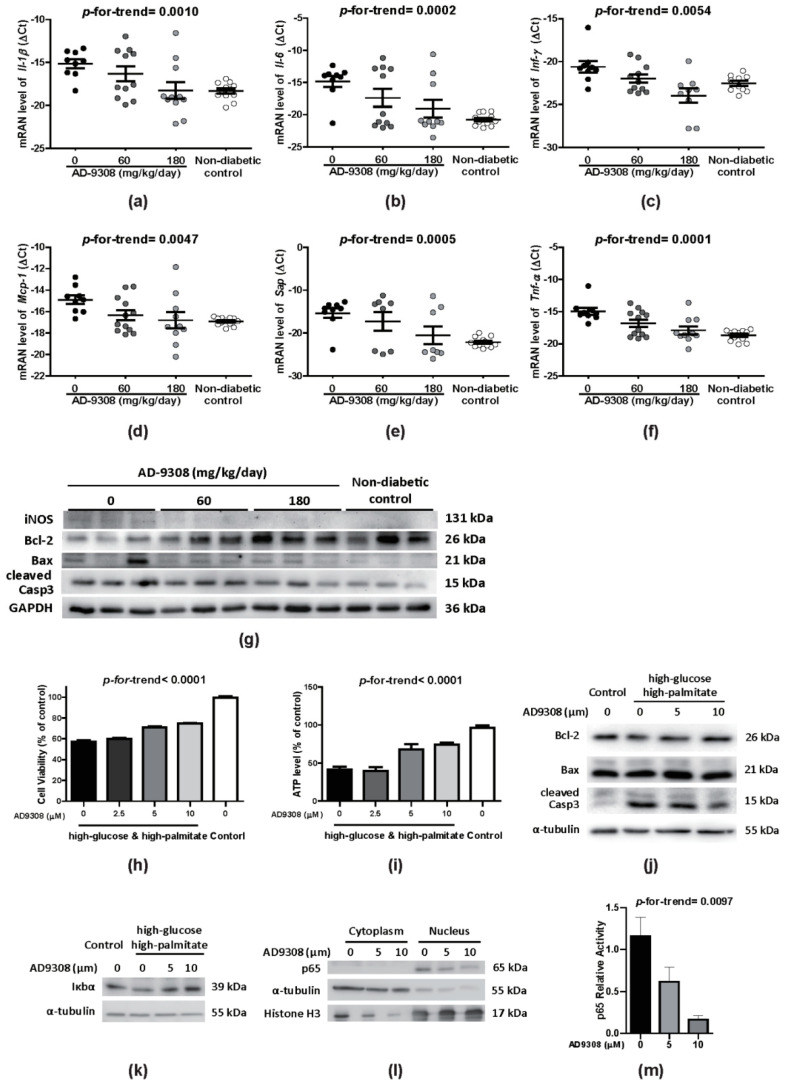
AD-9308 reduced inflammation in Streptozotocin (STZ)-induced diabetic mice and high-glucose and high-palmitate treated H9c2 cardiomyoblasts. In cardiac tissues from STZ-induced diabetic mice treated with 0, 60, 180 mg/kg/day of AD-9308 by oral gavages and non-diabetic control mice, the levels of mRNA transcripts of genes coding inflammation markers, including (**a**) interleukin 1β(*Il-1β*), (**b**) *Il-6*, (**c**) interferon gamma (*Infγ*), (**d**) monocyte chemoattractant protein 1 (*Mcp-1*), (**e**) serum amyloid P-component (*Sap*), (**f**) tumor necrosis factor-α (*Tnf-α*), and (**g**) the immunoblottings of inflammation marker, inducible nitric oxide synthase (iNOS), and apoptosis markers, including B-cell lymphoma 2 (Bcl-2), Bcl-2-associated X protein (Bax) and cleaved caspase 3 (Casp3) protein expressions were detected. Data are presented as mean ± SEM (*n* = 14–16 per group). Relative (**h**) cell survival and (**i**) ATP production were measuredin 72-hhigh-glucose and high-palmitate incubated H9c2 cells without or with 1-h AD-9308 pretreatment. Immunoblottingsof (**j**) apoptotic markers (Bcl-2, Bax and cleaved Casp3), (**k**) inflammatory marker (Iκbα) and (**l**) fractionated cytosolic and nuclear proteins of NF-κB p65 (using α-Tubulin and Histone H3 as cytosolic and nuclear internal controls, respectively) were measured in 72-hhigh-glucose and high-palmitate incubated H9c2 cells without or with 24-h AD-9308 pretreatment. (**m**) The mean NF-κB p65 reporter activity was measured in 72-hhigh-glucose and high-palmitate incubated H9c2 cells without or with 1-h AD-9308 pretreatment. Data are presented as mean ± SEM (*n* = 3–4 per group). *p*-for-trend was used to test the linear trend.

**Figure 7 antioxidants-10-00450-f007:**
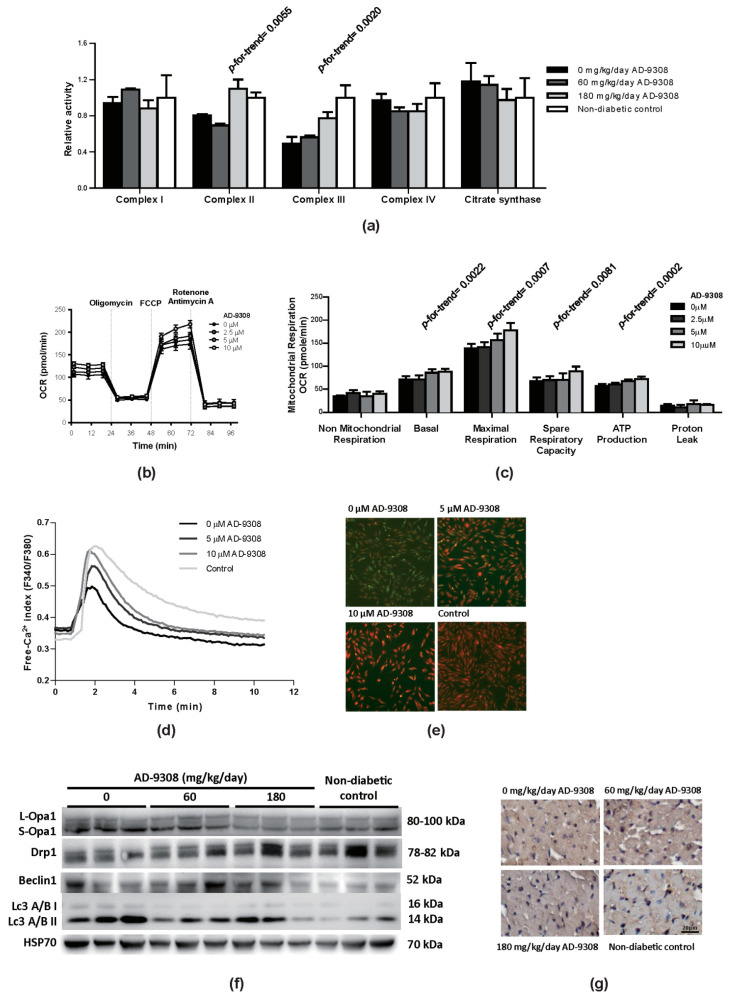
AD-9308 improved mitochondrial functions and calcium homeostasis in streptozotocin (STZ)-induced diabetic mice and high-glucose and high-palmitate treated H9c2 cardiomyoblasts. (**a**) The activities of mitochondrial electron transfer chain complex were measured in cardiac tissue from STZ-induced diabetic mice treated with 0, 60, 180 mg/kg/day of AD-9308 by oral gavages and non-diabetic mice; (**b**) Mitochondrial respiration and (**c**) mitochondrial oxygen consumption rate in high-glucose and high-palmitate cultured H9c2 cells without or with AD-9308 treatment were measured. Data are presented as mean ± SEM (*n* = 3–6 per group). *p*-for-trend was used to test the linear trend. (**d**) Fluorescence spectra from Fura-2-AM loaded H9c2 cells were analyzed under high-glucose and high-palmitate medium-cultured condition (pretreated with 0, 5, and 10 μM AD-9308) and normal condition (control), and (**e**) represent Fura2-AM fluorescent images of free Ca^2+^ release from sacroendoplamic reticulum. (**f**) Immunoblottings of mitochondrial dynamic regulators, including long-form Optic atrophy 1 (L-Opa1), short-form Optic atrophy 1 (S-Opa1) and dynamin-related protein 1 (Drp1), and autophagy regulators, including Beclin1and Microtubule-associated proteins 1A/1B light chain 3A and 3B (LC3A/B) protein I and II, protein expressions and (**g**) representative immunohistochemical staining of LC3 were detected in cardiac tissue from STZ-induced diabetic mice treated with 0, 60, 180 mg/kg/day of AD-9308 by oral gavages and non-diabetic control mice.

**Figure 8 antioxidants-10-00450-f008:**
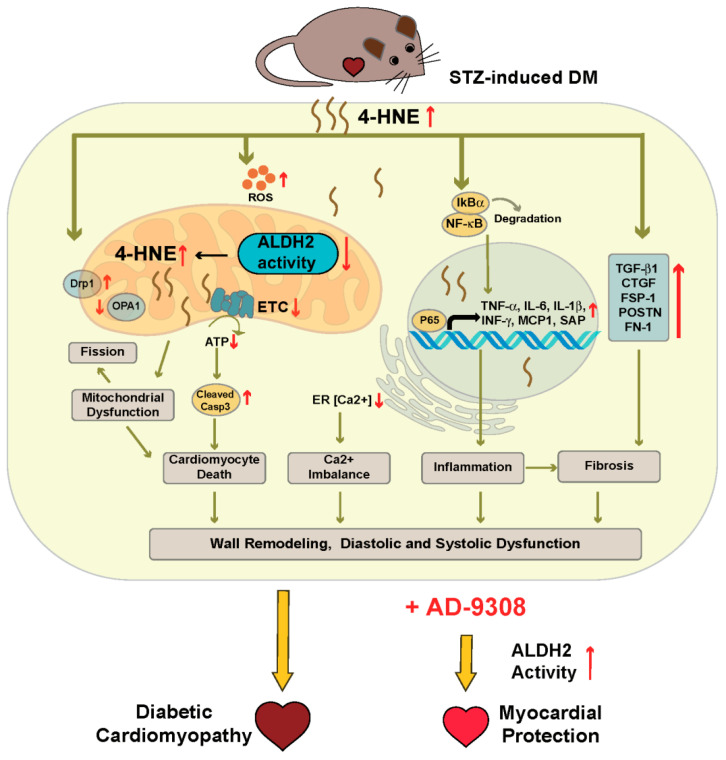
Schematic diagram depicting the effect of AD-9308 treatment on diabetic mice.

## Data Availability

The data presented in this study are available on request from the corresponding author.

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
