# Peer review of "A Novel ALDH2 Activator AD-9308 Improves Diastolic and Systolic Myocardial Functions in Streptozotocin-Induced Diabetic Mice"

_antioxidants, 2021, doi:10.3390/antiox10030450_

Round 1

Reviewer 1 Report

This is a well done study and it is well presented.

In reading the paper and the DISCUSSION it would appear that certain parts of the study have been done before by other groups. How much of this study is new and original? Is all the data on AD 9308 new or is it replicating what others have done? 

Also in humans  - how would this protein be administered? PO or IM or IV?

Otherwise this is a good, well done study.

Author Response

  1. This is a well done study and it is well presented.

Answer: We thank the reviewer for the encouragement.

  1. In reading the paper and the DISCUSSION it would appear that certain parts of the study have been done before by other groups. How much of this study is new and original? Is all the data on AD 9308 new or is it replicating what others have done?

Answer: Prior studies have showed the relationship between ALDH2 activity and heart failure; however there is no effective treatment provided. In this study, we for the first time show that activation of ALDH2 by a novel, highly potent, and selective ALDH2 activator, which has passed phase 1 human clinical trial with acceptable safety and tolerability could potentially benefit the diabetic cardiomyopathy. All the data of AD9308 is new. We have clarified these issues in the revised discussion and added statement stressing the novelty about AD-9308 as follows:

“However, few of them were successfully translated into clinically approved therapy. AD-9308 is a novel, highly potent, selective, and water-soluble ALDH2 activator, which has passed phase 1 human clinical trials with acceptable safety and tolerability in healthy subjects. Our findings may be translated into clinical trials soon.”

  1. Also in humans - how would this protein be administered? PO or IM or IV?

Answer: AD-9308 a water-soluble small-molecule ALDH2 activator given by oral route. We have added this information in the revised discussion.

  1. Otherwise this is a good, well done study.

Answer: We thank the reviewer’s positive comments.

Reviewer 2 Report

This manuscript submitted by Lee et al described the effect of AD-9308 on cardiomyopathy. This has some interesting results, however the experiments are not performed in sufficient depth and stringency to warrant the authors' conclusion. Moreover, some of findings are strongly over-interpreted. It is necessary to refine the experimental design more and to output persuasive data and logical and detail explanation.

Comments

  1. The title is misleading in this context.
  2. Figure 1a: “for 7 days” in the text, while “for 2 days” in the figure. Which one is correct?
  3. The number of dots in each figure differs from the number described in its legend. Why?
  4. Figure 3h: “no significant difference in myocardial fiber size” in the text. You should show the quantitative results of fiber size in mice. Figure legend (i)→(h)
  5. Figure 6: apoptosis and inflammation. I strongly recommend you show immunostaining data rather than data using rat cell line.
  6. Figure 6m: I cannot find.
  7. Figure 7f: more direct evidences are needed. You should additionally show immunostaining data.
  8. Figure legends: you need to clarify exactly what you did.
  9. Discussion section is too long. The authors should focus attention on the main points.

Author Response

This manuscript submitted by Lee et al described the effect of AD-9308 on cardiomyopathy. This has some interesting results, however the experiments are not performed in sufficient depth and stringency to warrant the authors' conclusion. Moreover, some of findings are strongly over-interpreted. It is necessary to refine the experimental design more and to output persuasive data and logical and detail explanation.

Answer: We thank the reviewer’s constructive criticism and suggestions. We have done additional experiments to improve this study. We do revise in our revised manuscript on study design and the logical explanation.

Comments:

  1. The title is misleading in this context.

Answer: Thanks for the comment. We have changed the title to “A novel ALDH2 activator AD-9308 improves diastolic and systolic myocardial functions in streptozotocin-induced diabetic mice”.

  1. Figure 1a: “for 7 days” in the text, while “for 2 days” in the figure. Which one is correct?

Answer: We thank the reviewer’s careful reading. “For 7 days” is correct. In the Figure 1a, 7 days is indicated as one week.

  1. The number of dots in each figure differs from the number described in its legend. Why?

Answer: The reason of difference between dots and number described is due to difficult to measure E and A wave in some mice. We have added this information in the “Materials and Methods” as follows:

“Each group included 14-16 mice. However, E wave and A wave of some mice are missing due to technical difficulty.”

  1. Figure 3h: “no significant difference in myocardial fiber size” in the text. You should show the quantitative results of fiber size in mice. Figure legend (i)→(h).

Answer: We thank reviewer for the constructive suggestion. We have added the quantitative estimation of myocardial fiber size, shown in the Figure S2. The Figure legend has been corrected.

  1. Figure 6: apoptosis and inflammation. I strongly recommend you show immunostaining data rather than data using rat cell line.

Answer: We have performed immunoblottings of inflammation and apoptosis makers in addition to RT-qPCR, as shown in Figure 6a-6g. Indeed, the data obtained from cell line studies add additional evidence.

  1. Figure 6m: I cannot find.

Answer: We thank the reviewers’ careful reading and have corrected Figure 6 in the revised manuscript.

  1. Figure 7f: more direct evidences are needed. You should additionally show immunostaining data.

Answer: We thank the reviewers’ comments. We have performed additional IHC for LC3 II, an autophagy marker in Figure 7g of the revised manuscript.

  1. Figure legends: you need to clarify exactly what you did.

Answer: We thank the reviewer for this constructive comment. We have extensively modified the figure legends in the revised manuscript.

  1. Discussion section is too long. The authors should focus attention on the main points.

Answer: We have shortened the Discussion according to reviewer’s critical comments. Please find these changes in the revised manuscript.

Reviewer 3 Report

Lee et al report a mouse study that investigated the effect of ALDH2 activator on diabetic cardiac dysfunction. They showed that AD-9308 given at 10 weeks of age after diabetes induction in mice prevents cardiac dysfunction, suppresses fibrotic marker increase, and improves mitochondrial function in a dose-dependent manner. The results are generally consistent and show promising effect of AD-9308 in preventing diabetic cardiac dysfunction in this model. Some additional data may improve the manuscript. Following are my specific comments.

Major
Although ALDH2 activity is examined, ALDH2 expression is not studied. Whether its expression is altered is important for chronic diseases.
The statistic method described is inappropriate or incorrect. Cochran-Armitage test should be applied to the 2 variable data and not for continuous parameters. Some of the figs with bar graphs should indicate the number of animals/samples or use dot data.
ALDH2 activation is not cardiac specific. Whether it was cardiac or systemic effect remains uncertain from the presented data. This should be examined or at least clearly stated as a limitation. 
When was the glucose level (supplement data) measured? 
LV mass is usually increased in diabetic heart. Decrease may be associated with lower body weight. Did the authors try to adjust?

Minor
Fig 4 legend says (C) isheme oxygenase-1, but the graph says ALDH2 activity
In discussion, “none of them were successfully translated into clinically …” is not true. Chronic metformin is clinically used.

Author Response

Lee et al report a mouse study that investigated the effect of ALDH2 activator on diabetic cardiac dysfunction. They showed that AD-9308 given at 10 weeks of age after diabetes induction in mice prevents cardiac dysfunction, suppresses fibrotic marker increase, and improves mitochondrial function in a dose-dependent manner. The results are generally consistent and show promising effect of AD-9308 in preventing diabetic cardiac dysfunction in this model. Some additional data may improve the manuscript. Following are my specific comments.

Answer: We thank reviewer for the positive comment and encouragement.

Major

  1. Although ALDH2 activity is examined, ALDH2 expression is not studied. Whether its expression is altered is important for chronic diseases.

Answer: We thank reviewer for the comment and suggestion. We have added the immunoblots of ALDH2 in revised Figure S3.

  1. The statistic method described is inappropriate or incorrect. Cochran-Armitage test should be applied to the 2 variable data and not for continuous parameters. Some of the figs with bar graphs should indicate the number of animals/samples or use dot data.

Answer: We thank reviewer for the critical comment and suggestion. We have modified the term to” ANOVA with trend test” in the revised “Materials and Methods” and added number of animals/samples in the figure legends.

  1. ALDH2 activation is not cardiac specific. Whether it was cardiac or systemic effect remains uncertain from the presented data. This should be examined or at least clearly stated as a limitation.

Answer: We agree with the reviewer and add a paragraph pointing out this study limitation in the revised discussion, as follows:

“Our study has some limitations. First, ALDH2 is ubiquitously expressed. However, we only explore myocardium, leaving other organs unexplored. Indeed, 4-HNE has been shown to oxidize LDL and promote oxLDL accumulation in atherosclerotic plaques in vessels [81, 82]. 4-HNE also induced ER stress in endothelial cells [83]. ER stress exerts a negative effect on endothelial cell stability [84]. Endothelial dysfunction may increase arterial stiffness and resistance and cause myocardial hypertrophy. And, therefore the effect of ALDH2 activation might be through a secondary effect from other cells in addition to the myocardium and the cardiac myocytes as characterized in this study.”

  1. When was the glucose level (supplement data) measured?

Answer: We measured blood glucose levels at the age of 5 months. We have added in the timing in the legend of Figure S1.

  1. LV mass is usually increased in diabetic heart. Decrease may be associated with lower body weight. Did the authors try to adjust?

Answer: We thank reviewer for the comment. Although the phenotypes and underlying mechanisms of diabetic cardiomyopathy mostly overlap, some distinct alterations exist between type 1 and type 2 DM in humans. Prior studies on STZ-induced diabetic mice, mimicking human type 1 diabetes, showed that estimated LV mass were decreased as compared to control mice, see following references (1-4). More importantly, we had detected a decline in the LV mass of STZ-induced diabetic mice at all three investigated time points versus the respective controls (Fig. 1c).

But when we adjusted the LV mass with body weight, we found that there was no significant difference among groups, now shown in Figure S1. F.

References:

  1. J. Mol. Sci.2016, 17(12), 2136; https://doi.org/10.3390/ijms17122136
  2. J. Mol. Sci. 2018, 19(10), 3094; https://doi.org/10.3390/ijms19103094.

  Table 1

  1. Scientific Reports, 27 Feb 2020, 10(1):3629; DOI: 10.1038/s41598-020-60594-2.
  2. J Clin Invest. 2021;131(2):e95747. https://doi.org/10.1172/JCI95747.

Minor

  1. Fig 4 legend says (C) isheme oxygenase-1, but the graph says ALDH2 activity

In discussion, “none of them were successfully translated into clinically …” is not true. Chronic metformin is clinically used.

Answer: We thank reviewer for the comments. We have corrected the legend as “(a) serum 4-HNE levels, (b) 4-HNE protein adducts, (c) ALDH2 activities and (d) hemeoxygenase-1 (Ho-1) protein expressions of cardiac tissues were measured in STZ-induced diabetic mice treated with 0, 60, 180 mg/kg/day of AD-9308 by oral gavages and non-diabetic control mice.” And, in discussion, we modified the sentence to “few of them were successfully translated into clinically …” according to reviewer’s comment and suggestion.

Round 2

Reviewer 2 Report

The manuscript has been revised well. I would recommend it for acceptance after the minor points.

Minor point:

You should add a description of how to measure myocardial fiber size.

“myocardial fiber size” in the text, “cardiomyocyte size” in legend of Figure S2. The terms are different in each section.

Author Response

The manuscript has been revised well. I would recommend it for acceptance after the minor points.

Answer: We thank reviewer for the positive comment and encouragement.

Minor point:

  1. You should add a description of how to measure myocardial fiber size.

Answer: We thank the reviewer’s careful reading. We have added the method of estimation of cardiomyocyte size in Supplementary Materials and Methods.

  1. “myocardial fiber size” in the text, “cardiomyocyte size” in legend of Figure S2. The terms are different in each section.

Answer: We thank the reviewer’s careful reading. We have change “myocardial fiber size” in the text into “cardiomyocyte size”.